# scVGATAE: A Variational Graph Attentional Autoencoder Model for Clustering Single-Cell RNA-seq Data

**DOI:** 10.3390/biology13090713

**Published:** 2024-09-11

**Authors:** Lijun Liu, Xiaoyang Wu, Jun Yu, Yuduo Zhang, Kaixing Niu, Anli Yu

**Affiliations:** School of Science, Dalian Minzu University, Dalian 116600, China; manopt@163.com (L.L.); wuxiaoyangzhuxiaoming@outlook.com (X.W.); zhangyuduo@dlnu.edu.cn (Y.Z.); kx65a4@outlook.com (K.N.); alyu1230@163.com (A.Y.)

**Keywords:** scRNA-seq, unsupervised clustering, variational graph autoencoder, graph attention networks

## Abstract

**Simple Summary:**

Due to the rapid development of single-cell RNA sequencing technology, the volume of single-cell RNA sequencing data has grown exponentially. Traditional clustering methods have proven increasingly difficult to cluster this large-scale and highly complex single-cell RNA sequencing data. Although many unsupervised clustering methods based on deep neural networks have been developed to cluster cell subpopulations, these methods are complex in models and poor in scalability. In this paper, we propose a novel clustering method for single-cell RNA sequencing, which successfully combines the advantages of these two clustering models, maintaining high clustering performance while also preserving the stable computational efficiency of traditional clustering methods. Experiments conducted on nine public datasets have demonstrated that our proposed novel clustering method for single-cell RNA sequencing outperforms both classic and state-of-the-art clustering methods.

**Abstract:**

Single-cell RNA sequencing (scRNA-seq) is now a successful technology for identifying cell heterogeneity, revealing new cell subpopulations, and predicting developmental trajectories. A crucial component in scRNA-seq is the precise identification of cell subsets. Although many unsupervised clustering methods have been developed for clustering cell subpopulations, the performance of these methods is prone to be affected by dropout, high dimensionality, and technical noise. Additionally, most existing methods are time-consuming and fail to fully consider the potential correlations between cells. In this paper, we propose a novel unsupervised clustering method called scVGATAE (Single-cell Variational Graph Attention Autoencoder) for scRNA-seq data. This method constructs a reliable cell graph through network denoising, utilizes a novel variational graph autoencoder model integrated with graph attention networks to aggregate neighbor information and learn the distribution of the low-dimensional representations of cells, and adaptively determines the model training iterations for various datasets. Finally, the obtained low-dimensional representations of cells are clustered using kmeans. Experiments on nine public datasets show that scVGATAE outperforms classical and state-of-the-art clustering methods.

## 1. Introduction

Single-cell RNA sequencing (scRNA-seq) technology has revolutionized research progress in the field of biology. It can analyze gene expression patterns at the single-cell level, providing an unprecedented perspective for exploring cell heterogeneity and understanding life processes and disease mechanisms [1]. Through scRNA-seq technology, researchers can obtain transcriptome information of individual cells, and then depict complex cell graphs, providing an important basis for precision medicine and drug development [2,3]. The main steps of scRNA-seq analysis usually include data quality control, gene expression quantification, feature extraction, dimensionality reduction analysis, and cell clustering. Among them, dimensionality reduction analysis and cell clustering undoubtedly play a crucial role, as they are of decisive significance for accurately inferring cell development trajectories and effectively identifying rare cell clusters [4].

However, due to the inherent high dimensionality, technical noise, common dropout events, and batch effects of scRNA-seq data are inevitable [5]. Traditional clustering methods often face multiple challenges such as pattern recognition difficulties, significant noise interference, severe information loss, and unstable results when processing these data, which greatly limits their application effectiveness in single-cell analysis. Therefore, developing efficient and accurate scRNA-seq data clustering methods has become a research hotspot in the current field of bioinformatics.

Cell clustering analysis primarily relies on dimensionality reduction techniques and traditional clustering methods. Given the high-dimensional nature of scRNA-seq data, the primary objective is to derive a low-dimensional representation. Common dimensionality reduction techniques include principal component analysis (PCA), t-distributed stochastic neighborhood embedding (t-SNE) [6], diffusion maps, and uniform manifold approximation and projection (UMAP) [7]. Subsequently, traditional clustering methods, such as k-means and hierarchical clustering, are applied to the learned low-dimensional representations of cells. These methods cluster cells based on their similarity. While these traditional clustering methods are straightforward, efficient, and highly interpretable, they often fall short when dealing with high-dimensional and complex data.

To enhance clustering performance on scRNA-seq data, many methods integrate various similarity metrics and clustering outcomes. For instance, SC3 [8] employs diverse approaches to calculate multiple cell similarity matrices in parallel, applies k-means to these matrices, and subsequently applies hierarchical clustering to a consensus matrix formed from the results. SAME [9], a hybrid model-based clustering technique, seeks to derive diverse solutions from various clustering methods and generates an enhanced ensemble solution by selecting the most diverse subset. RCSL [10] assesses global and local cell relationships to differentiate cell types, evaluates global similarity using Spearman’s rank correlation, and learns neighborhood representations for local similarity. These are then linearly combined to produce block diagonal matrices, leading to the final clustering results. While these methods enhance the robustness of clustering results, reflecting cell similarities and differences more accurately, pairwise similarity only captures superficial relationships among cells and fails to capture the deep complex underlying relationships, leading to clustering results that may be insufficiently accurate or comprehensive.

In contrast, deep neural networks possess robust feature learning capabilities, enabling them to more effectively utilize information from scRNA-seq data to enhance clustering accuracy. Examples include scGMAI [11], DCA [12], scDeepCluster [13], and scVAE [14]. Specifically, scGMAI employs autoencoders to concurrently learn feature representations and cluster assignments, reconstructs gene expression values using stacked autoencoders, selects significant independent features via FastICA, and clusters cells using Gaussian mixture models. The DCA method substitutes the conventional mean squared error (MSE) loss function with a loss function based on the zero-inflated negative binomial (ZINB) model to more accurately characterize scRNA-seq data. scDeepCluster also employs an autoencoder based on the ZINB model to enhance the learning of latent features for subsequent clustering. The scVAE method utilizes a deep variational autoencoder for clustering scRNA-seq data.

Although these clustering methods based on deep neural networks have significantly enhanced the accuracy of clustering results, they exhibit notable shortcomings. Firstly, they typically concentrate on learning the intrinsic features of cells, overlooking the structural relationships among them. Indeed, this structural information can effectively unveil the latent similarities among cells and more precisely capture their actual relationships, leading to more accurate clustering outcomes [15]. To naturally embed both the cell feature expression information and the structural relationships among cells into the learned low-dimensional representations of cells, the clustering analysis method based on Graph Neural Network (GNN) for scRNA-seq data has emerged as the need has arisen. These methods can not only deeply explore the intrinsic structure of single-cell data, but also reveal the complex interactions between cells. Several methods, including CellVGAE [16] and scVGAE, utilize the GNN structure for cell clustering. CellVGAE uses the structural relationships between cells (such as KNN) and gene expression values as input features of the model, integrates graph attention mechanism into variational graph autoencoder, learns high-quality low dimensional representations of cells, and performs k-means clustering on the obtained low dimensional representations of cells. scVGAE integrates Graph Convolutional Networks (GCNs) into the Variational Autoencoder framework and utilizes the ZINB loss function to effectively address missing events in scRNA-seq data. Finally, the interpolated cell expression matrix is clustered. However, since VGAE merely employs the inner product of low-dimensional cell representations as a decoder to reconstruct the original graph and relies on reconstruction loss along with KL divergence regularization as loss functions to optimize model parameters, it fails to accurately capture the complexity and diversity of cellular structural relationships. It exhibits sensitivity to noise and outliers, and the inflexible reconstruction approach severely limits the learning of low-dimensional cell representations. There is a need to employ more sophisticated decoder structures to more effectively capture the critical information and structural features among cells. Recent methods, such as scGNN2.0 [17] and scGAC [18], have employed graph attention autoencoders for the clustering analysis of scRNA-seq data, achieving favorable clustering results. scGNN2.0 leverages GAT and a multimodal autoencoder to formulate and aggregate relationships between cells. scGAC employs graph autoencoders with attention mechanisms to learn cluster-friendly feature representations and utilizes self-optimizing clustering to enhance clustering performance. Nonetheless, these graph autoencoder models based entirely on GAT entail substantial computational costs and memory footprint, and they lack sufficient convenience and scalability for massive and high-dimensional scRNA-seq data.

To overcome these challenges, we propose a novel clustering method for scRNA-seq data named ‘scVGATAE’ (Single-cell Variational Graph Attention Autoencoder). Firstly, a cell graph is constructed using the Pearson correlation coefficient. Due to the characteristics of scRNA-seq data, scVGATAE employs the diffusion-based denoising strategy of NE to capture the potential similarities among cells from the high-order structure of the cell graph [19]. Based on the cell graph denoised by NE, scVGATAE learns the low-dimensional representations of cells through a variational graph autoencoder integrated with graph attention networks. Specifically, the inner product decoder of VGAE is replaced with two stacked graph attention layers, and a custom loss function is utilized to optimize the model parameters. Subsequently, we adopted an adaptive strategy to select appropriate training times on different datasets and applied k-means clustering analysis. To benchmark the performance of the proposed method, we compared scVGATAE with the six most popular scRNA-seq clustering methods on nine real and effective scRNA-seq datasets. The results showed that scVGATAE not only significantly improved the clustering performance of scRNA-seq data, but also maintained relatively stable running time at various cell orders of magnitude.

## 2. Materials and Methods

### 2.1. Architecture of scVGTAE

The initial input of scVGATAE is the raw UMI count matrix of sRNA seq, where each row represents a single cell and each column represents a gene. Each element in the matrix represents the count of Unique Molecular Identifiers (UMIs) for a specific gene in the corresponding cell. However, due to various technical noises and variations that may exist during the experimental process, the UMI counting matrix requires quality control and filtering, as well as pre-processing operations such as matrix normalization and logarithmization, to obtain the preprocessed matrix $X_{n\times m}$. Secondly, scVGATAE calculates the similarity matrix $S_{n\times n}$ between cells based on the Pearson correlation coefficient and denoises it through network enhancement (NE) to obtain matrix $E_{n\times n}$. Based on matrix $E_{n\times n}$, perform the nearest k-nearest neighbor algorithm and normalize the resulting matrix to obtain cell graph $A_{n\times n}$. The above is the way to obtain cell graph data among the multimodal input data for the variational graph attention autoencoder. The other cell feature expression matrix $Y_{n\times d}$ is obtained by performing PCA dimensionality reduction on the $X_{n\times m}$ matrix. The encoder of the variational graph attention autoencoder (graph convolutional network) takes the cell feature expression matrix $Y_{n\times d}$ and cell graph $A_{n\times n}$ as multimodal inputs, learns the mean μ and variance σ of the low dimensional vector representation of cells, and then uses the reparameterization technique to sample the variable $Z_{n\times b}$ from $N\left(\mu ,\sigma^{2}\right)$. Then, the decoder (graph attention network) is used to reconstruct the cell feature expression matrix $Y_{n\times d}^{\prime }$. The loss function comprises two parts: the mean absolute error between the reconstructed cell feature expression matrix $Y_{n\times d}^{\prime }$ and the original cell feature expression matrix $Y_{n\times d}$ as well as the divergence between the distribution $N\left(\mu ,\sigma^{2}\right)$ of the cell’s low-dimensional representation vectors and a normal distribution. Finally, repeat the encoding and decoding operations until the stop condition is reached. The final low dimensional representation vector $Z_{n\times b}$ of cells is obtained through the trained model, and the k-means clustering method is used for cell clustering. The number of cell clusters in the k-means clustering is determined by the Leiden diagram clustering method (based on optimized modular community detection). Firstly, the original cell feature expression matrix $Y_{n\times d}$ is subjected to cell–cell neighborhood graph calculation. Then, the obtained neighborhood graph is subjected to the Leiden clustering algorithm, and finally, the number of cell clusters is obtained. Figure 1 shows the main architecture of scVGATAE.

### 2.2. Construction of a Cell Graph Based on NE Denoising

In order to share information between similar cells and learn a more friendly distribution of low dimensional cell feature vectors, the variational graph attention autoencoder needs to construct a graph structure that can more accurately capture the similarity between cells, in order to obtain a highly reliable cell map.

Firstly, the Pearson correlation coefficient is used to construct the initial cell similarity matrix $S_{n\times n}$ to evaluate the degree of similarity in feature expression between different cells. The formula is as follows
(1)$$S_{ij}=\frac{\sum_{i=1,j=1}^{n}\left(x_{i}-\bar{x}\right)\left(x_{j}-\bar{x}\right)}{\sqrt{\sum_{i=1}^{n}{\left(x_{i}-\bar{x}\right)}^{2}}\sqrt{\sum_{j=1}^{n}{\left(x_{j}-\bar{x}\right)}^{2}}},$$

However, scRNA-seq data may contain missing values or outliers, which reduces the reliability of the Pearson correlation coefficients between pairs of cells. This can mislead the sharing of information between similar cells, ultimately leading to suboptimal cell clustering results.

We adopt Network Enhancement (NE) to improve the signal-to-noise ratio of $S_{n\times n}$. NE has designed an effective diffusion denoising strategy that utilizes random walks with a length of no more than 3 and information regularization techniques to optimize the network structure. Among them, a random walk with a length not exceeding 3 means starting from one node and randomly selecting its adjacent nodes for up to three jumps. Through this method, local structural information in the network can be captured, especially the relationships between high weight edges. This local information is crucial for denoising and optimizing network structures. Secondly, information regularization is used to ensure that the diffusion process does not excessively distort the original network structure, while effectively eliminating noise. Finally, NE updates the initial cell similarity matrix $S_{n\times n}$ using a diffusion based method to obtain a cell enhanced similarity matrix ${\hat{S}}_{n\times n}$.

Specifically, first, based on the initial cell similarity matrix $S_{n\times n}$, calculate the transition matrix P and the local network matrix T, which are, respectively, represented as follows:(2)$$P_{ij}=\frac{S_{ij}}{\sum_{k\in N_{i}}S_{ik}},j\in N_{i},$$
(3)$$T_{ij}=\sum_{k=1}^{n}\frac{P_{ik}P_{jk}}{\sum_{v=1}^{n}P_{vk}},$$
Among them, $N_{i}$ represents the set of k neighboring nodes closest to node i; After extensive experimental verification, this paper has k=10.

Subsequently, NE uses the local network matrix T to update the initial cell similarity matrix $S_{n\times n}$, which is the NE diffusion process. The process is as follows:(4)$$\hat{S}=\alpha T\times S\times T+\left(1-\alpha \right)T,$$
Among them, α is the regularization parameter.

Finally, obtain the denoising similarity matrix $E_{n\times n}$ according to the following formula.
(5)$$E_{ij}=\left\{\begin{array}{c} S_{ij}\;\;\;\;if\;{\hat{S}}_{ij}\geq t\\ 0\;\;\;\;\;otherwise \end{array}\right.,$$
where *t* is a predefined threshold. If ${\hat{S}}_{ij}$ is too small (less than *t*), $S_{ij}$ will be considered as a noise edge and $E_{ij}$ will take on a value of zero. Then, we perform the nearest k-nearest neighbor algorithm and normalization on matrix $E_{ij}$ and finally obtain cell graph $A_{n\times n}$.

### 2.3. Variational Graph Attention Autoencoder

Inspired by the Variational Graph Autoencoder (VGAE) [20] and Graph Attention Network (GAT) [21], we designed a Variational Graph Attention Autoencoder. Firstly, the cell graph $A_{n\times n}$ and cell feature expression matrix $Y_{n\times d}$ are input. An encoder, composed of two graph convolutional network layers, is used to learn the mean μ and variance σ of the low-dimensional vector representations of the cells. Subsequently, the reparameterization trick is employed to sample the low-dimensional vector representations $Z_{n\times b}$ of the cells from $N\left(\mu ,\sigma \right)$. Finally, a decoder with two stacked graph attention network layers is used to reconstruct the cell feature expression matrix $Y_{n\times d}^{\prime }$ from the low-dimensional representation $Z_{n\times b}$. The variational graph attention autoencoder learns the network parameters through a custom loss function, embedding the network topology information of the cell graph into the low-dimensional $Z_{n\times b}$ representations of the cells.

#### 2.3.1. Encoder Based on the Graph Convolutional Network 

The encoder is a stacked two-layer graph convolutional network
(6)$$q\left(Z|Y,A\right)=\prod_{i=1}^{n}q\left(Z_{i}|Y,A\right),$$
where $q\left(Z_{i}|Y,A\right)=N\left(z_{i}|\mu_{i},diag(\sigma_{i}^{2})\right)$. μ is the mean of the low dimensional vector representation $Z_{n\times b}$ of the cell to be learned,$\;\mu ={GCN}_{\mu }\left(Y,A\right)$. σ is the variance of the low dimensional representation $Z_{n\times b}$ of the cell to be learned, and $log\sigma ={GCN}_{\sigma }\left(Y,A\right)$.

The definition of a two-layer graph convolutional network is as follows:(7)$${GCN}_{\mu }\left(Y,A\right)=\tilde{A}ReLU\left(\tilde{A}XW_{0}\right)W_{1},$$
(8)$${GCN}_{\sigma }\left(Y,A\right)=\tilde{A}ReLU\left(\tilde{A}XW_{0}\right)W_{2},$$
where $\tilde{A}=D^{-\frac{1}{2}}AD^{-\frac{1}{2}}$. ${GCN}_{\mu }$ and ${GCN}_{\sigma }$ share the first layer of graph convolutional network, but have their own respective second layers of graph convolutional network.

The reparameterization trick helps us sample from the learned distribution of low-dimensional cell representation vectors $N\left(\mu ,\sigma^{2}\right)$ while preserving gradient information, thereby optimizing the model parameters. The formula is as follows:(9)Z=μ+ε×σ,
where ε is sampled from the standard normal distribution $N\left(0,1\right)$.

#### 2.3.2. Decoder Based on the Graph Attention Network 

The Variational Graph Autoencoder uses the inner product of the low-dimensional cell representations $Z_{n\times b}$ as the decoder to reconstruct the cell graph $A_{n\times n}$, as follows:(10)$$\tilde{A}=sigmoid\left(ZZ^{T}\right),$$

In fact, this decoder, which relies solely on the dot product operation of the low-dimensional cell representation matrix Z to reconstruct the cell graph, cannot accurately and flexibly reflect the true relationships between cells, limiting the reconstruction effect. This, in turn, affects the parameter optimization during backpropagation, ultimately restricting the richness and expressiveness of the distribution of the low-dimensional cell representations.

Therefore, we introduce graph attention networks into the variational graph autoencoder, using two stacked graph attention layers as the decoder. The graph attention layers learn the features of cells by aggregating the features of neighboring cells with different weights. Since the weights are dynamically assigned based on the similarity between a cell and its neighboring cells, it can fully capture the complexity and diversity of the data in the low-dimensional cell representations $Z_{n\times b}$ and flexibly embed topological information into the reconstructed cell feature matrix $Y_{n\times d}^{\prime }$, enhancing the reconstruction effect of the decoder.

Specifically, the computation of the graph attention network layer is divided into the following two steps:(1)Calculate the attention coefficient

For cell i, calculate the similarity coefficient between its neighbors j and itself one by one, using the formula
(11)$$e_{ij}=LeakyReLU\left(a^{T}\left[\left.Wh_{i}\right\|Wh_{j}\right]\right),j\in N_{i},$$
where $h_{i}\;and\;h_{j}$ are, respectively, the input features of cell i and cell j, $N_{i}$ is the set of neighbors of cell i, W is a learnable linear transformation matrix, ∥ is a concatenation operation, a is a learnable weight vector, and LeakyReLU is a nonlinear activation function.

Normalize the correlation coefficient using the softmax function to obtain the attention coefficient, with the formula
(12)$$a_{ij}=\frac{exp\left(e_{ij}\right)}{\sum_{k\in N_{i}}exp\left(e_{ik}\right)},$$
where $a_{ij}$ is the attention coefficient, indicating the importance of cell i to cell j.
(2)Weighted Sum

Based on the calculated attention coefficients, a weighted sum of the features of all neighboring cells is performed, and the formula is
(13)$$h_{i}^{\prime }=\sigma \left(\sum_{j\in N_{i}}a_{ij}Wh_{j}\right),$$
where σ is a nonlinear activation function. $h_{i}^{\prime }$ is a new feature of cell i.

#### 2.3.3. Composition of Loss Function 

The Mean Squared Error (MSE) loss function measures the error of the model by calculating the average of the squares of the differences between the reconstructed cell feature matrix $Y_{n\times d}^{\prime }$ and the original cell feature expression matrix $Y_{n\times d}$. The formula is as follows
(14)$$L_{MSE}=\frac{1}{n}{\left(Y-Y^{\prime }\right)}^{2},$$
where n is the total number of cells.

Similar to the Variational Graph Autoencoder, we add the KL divergence $\left(\left(\mu ,\sigma^{2}\right)\left\|\left(0,1\right)\right.\right)$ between each cell’s independent normal distribution and the standard normal distribution to the loss function, forcing each normal distribution to approximate the standard normal distribution
(15)$$L_{KL}=\frac{1}{2}\sum_{i=1}^{d}\left(\mu_{\left(i\right)}^{2}+\sigma_{\left(i\right)}^{2}-log\sigma_{\left(i\right)}^{2}-1\right),$$

Therefore, the loss function of the variational graph attention autoencoder is defined as:(16)$$L=\alpha L_{MSE}+\beta L_{KL},$$
where αandβ are coefficients that control the relative weights of two loss functions. 

#### 2.3.4. Adaptive Iterative Training

For different datasets, we provide a method to adaptively select the number of epochs for model training. The Silhouette coefficient [22] is calculated based on the low-dimensional cell representations and the predicted labels from k-means clustering to detect a declining trend in the clustering performance of the model. If a significant decrease in performance is detected, the training is stopped.

The Silhouette coefficient of a single cell is defined for each individual cell
(17)$$s=\frac{b-a}{max\left(a,b\right)},$$
where a represents the average distance between a cell and all other cells within the same cluster and b represents the average distance between that cell and all cells in the next closest cluster. The overall average of the Silhouette coefficients across all cells can be used as the Silhouette coefficient for the entire group of cells.

### 2.4. Datasets

The proposed scVGATAE unsupervised clustering method was evaluated for clustering performance on 9 real and effective scRNA-seq datasets, each containing cells labeled as previously or validated in previous studies. As shown in Table 1, the cell count of these datasets varies significantly, ranging from a few dozen cell samples to thousands of them, covering a wide range of data needs across various scales. Meanwhile, these data are sourced from diverse organizations and databases, fully demonstrating the richness and diversity of data sources. For example, the Adipose [23] dataset comes from the fat of a 36 year old male on the Human Cell Landscape (HCL) database, which contains 1372 cells and 8 cell types. However, the dataset Cerebellum [23] not only has a large scale, containing 7324 cells, but also contains an extremely rich variety of cell types, totaling 19 cell types. In addition, the real cell labels of these datasets are provided by the original database. 

In order to improve the generalization ability of the scVGATAE model, we adopted different data preprocessing strategies for datasets from different databases and scales. First, quality control and normalization were performed on the raw Unique Molecular Identifier (UMI) count matrix $R_{M\times N}\;$ of all scRNA-seq data to obtain the preprocessed matrix $X_{n\times m}$. We preprocessed the raw scRNA-seq data using the Python software package SCANPY (1.10.2) [29]. In order to provide a more reliable and accurate dataset for subsequent analysis, we filtered out low-quality and damaged cells, i.e., cells with less than 200 expressed genes and genes that may have been generated due to sequencing errors or sample contamination, i.e., genes expressed in less than three cells. Then, we normalized and log-transformed the data for the Biase, PBMC6k, Liver, Cerebellum, Heart, Kolodzisjczyk, and Baron datasets, while for the Goolam and Adipose datasets, we only performed log-transformation. After normalizing the dataset, we identified and filtered out highly variable genes to allow our model to focus on the important features that best capture the differences between cells, thereby enhancing the accuracy and interpretability of subsequent analyses. Finally, the filtered highly variable genes underwent scaling with a maximum value of 10.

### 2.5. Comparison of Clustering Performance

To further evaluate the clustering performance of scVGATAE, we compared it with six single-cell clustering methods. These methods can be divided into three categories, including traditional clustering approaches, deep neural network-based clustering approaches, and GNN (Graph Neural Network)-based clustering approaches.

Leiden [30] is a community detection algorithm encapsulated in the optimization module of scanpy;scVI [31] employs variational inference techniques to efficiently learn model parameters and approximate posterior distributions, enabling high-precision clustering analysis of single-cell RNA sequencing data;scDeepCluster [13] combines the Deep Embedding Clustering (DEC) algorithm in the field of image processing with the ZINB model to optimize clustering performance and feature learning;scASGC [32] is a cell clustering method based on an adaptive simplified graph convolutional network;scGNN2.0 [17] utilizes GAT and multimodal autoencoders to formulate and aggregate intercellular relationships;scGAC [18] applies a graph autoencoder that incorporates an attention mechanism to learn cluster-friendly feature representations, and employs self-optimization clustering to improve clustering;Among them, the parameters of the six scRNA seq clustering methods to be compared were set according to their tutorials.

To enhance the reliability and reproducibility of the experimental results, we independently performed five clustering analyses for each scRNA-seq clustering method on each dataset and calculated clustering performance evaluation metrics for each analysis. Finally, we took the average of these metrics to evaluate the clustering performance of each clustering method on that dataset.

### 2.6. Evaluation Metrics

In this paper, two widely used metrics are employed to evaluate the clustering performance of different algorithms: the Adjusted Rand Index (ARI) [33] and Normalized Mutual Information (NMI) [34].

ARI (Adjusted Rand Index) measures the similarity between the predicted cluster labels and the true cluster label assignments, with a value range of $\left[-1,1\right]$: negative values indicate poor labeling, higher values indicate higher clustering accuracy, and 1.0 represents a perfect matching score. The formula for calculating ARI is
(18)ARI=∑ijnij2−∑iai2∑jbj2/n212∑iai2+∑jbj2−∑iai2∑jbj2/n2,
where $n_{ij}$ is the number of cells shared between the cluster in the true cluster label assignment and cluster in the predicted cluster label assignment and n represents the total number of cells, $\;a_{i}=\sum_{j}n_{ij}$ and $b_{j}=\sum_{i}n_{ij}$.

NMI (Normalized Mutual Information) measures the consistency between the predicted cluster labels and the true cluster label assignments, ignoring arrangement. The formula for calculating NMI is
(19)$$NMI=\frac{2MI\left(U,V\right)}{H\left(U\right)+H\left(V\right)},$$
where U represents the predicted cluster labels,V represents the true cluster labels, $MI\left(U,V\right)$ denotes the mutual information between U and V, and $H\left(\cdot \right)$ is the entropy function.

## 3. Results

### 3.1. Compared with the Other Six Methods, scVGATAE Significantly Improves the Clustering Performance

The scVGATAE unsupervised scRNA-seq clustering method was compared in clustering performance with six widely popular single-cell clustering methods on nine real and effective scRNA-seq datasets. As shown in the last column of Figure 2, which depicts the average scores of the clustering metrics ARI and NMI for each method across nine datasets, scVGATAE achieves the highest scores for both clustering metrics. Notably, on six datasets, Biase, Goolam, Liver, Cerebellum, Heart, and Baron, scVGATAE attains the highest ARI and NMI scores, significantly outperforming other single-cell clustering methods. Slightly better are the PBMC6k, Adipose, and Kolodzisjczyk datasets. Specifically, on the Goolam dataset, scVGATAE demonstrated remarkable improvements over the suboptimal method scGAC, with a notable 10.1% increase in NMI and a substantial 23.2% enhancement in ARI. Similarly, on the mouse Baron dataset, scVGATAE surpassed scVI by 3.2% in NMI and outperformed scDeepCluster by 8.2% in ARI. In the Cerebellum dataset, compared to scDeepCluster, scVGATAE achieved a 13.6% lift in NMI, while over scGAC, it showed a significant 5% gain in ARI.

### 3.2. scVGATAE Learns Cluster-Friendly Embeddings for Cells

To demonstrate that the proposed scVGATAE clustering model not only significantly improves the clustering performance of scRNA-seq data but also improves the low dimensional representations of cells learned through it, which have friendly visual embeddings, firstly, UMAP is used to project the low dimensional representations of cells learned by scVGATAE into a two-dimensional space and visualize the two-dimensional cell representations. As shown in Figure 3, we selected two real datasets with different cell orders of magnitude and from different tissues, including human adipose cells (Figure 3a) and human cerebellar cells (Figure 3b). Here, scGNN2 represents scGNN2.0. 

In Figure 3a, we visually demonstrate the segregation of various cell types within a human adipose dataset through color coding. Notably, the scVGATAE method achieved remarkable and distinct separation of eight different cell subtypes, including stromal cells, mast cells, and two types of adipocytes expressing high levels of distinct genes (Adipocyte_FGR high and Adipocyte_SPP1 high). These cell clusters appear as compact and well-demarcated groups in the plot, underscoring scVGATAE’s prowess in differentiating subtle variations. In contrast, both Leiden and scVI methods exhibit considerable mixing of cell types in their UMAP embeddings, while scGNN2.0, scGAC, and scASGC, although achieving comparatively good cell embeddings, still display a minor blending of M2 macrophages and stromal cells. This comparison highlights the unique advantage of scVGATAE in single-cell RNA sequencing data analysis. Lastly, although scDeepCluster achieves good separation of distinct cell clusters, it results in overly distanced clusters, potentially obscuring the underlying correlations between them to some extent. 

As shown in Figure 3b, for the Cerebellum dataset, which contains over 7000 cells and 19 abundant cell clusters, scVGATAE still achieves a good separation of the dataset with clear boundaries. In comparison, scGNN2.0 and Leiden tend to fuse different cell clusters together. While scGAC, scASGC, and scVI are able to define relatively distinct cell cluster boundaries, they still face the issue of a small number of outliers being mixed in. In summary, compared to the other six methods, these either fail to clearly delineate the boundaries between cell clusters or compromise the intrinsic structural information of the cells’ low-dimensional embeddings during the learning process. However, the scVGATAE method demonstrates significant advantages: it not only precisely identifies diverse cell clusters and clearly defines the boundaries between them, but also successfully preserves the rich intrinsic structural information of the cell clusters.

### 3.3. Ablation Study

We set up four ablation study subjects of scVGATAE: scVGATAE_no_NE (scVGATAE without incorporating NE), scVGATAE_no_attention (a variational graph autoencoder without an attention network), scVGATAE_all_attention (a graph autoencoder solely based on an attention network), and scVGATAE_no_self-adaption-epochs (without self-adaptive model training epochs, using a default of 180 epochs).

As shown in Figure 4, scVGATAE achieves the best clustering performance compared to its four ablation study subjects. The results indicate that the NE (Network Enhancement) denoising module, the variational graph autoencoder model updated based on graph attention networks, and the adaptive adjustment of model training iterations all play significant roles in enhancing the overall performance of the model.

### 3.4. Scalability and Efficiency

To validate the scalability and model execution efficiency of scVGATAE, we compared its runtime with six other scRNA-seq clustering methods. As shown in Figure 5, the runtimes of scGNN2.0, scGAC, and scDeepCluster significantly increase with the increase in the magnitude of cell numbers, whereas scVGATAE maintains a relatively stable runtime across various cell magnitudes. This demonstrates that scVGATAE not only possesses excellent scalability, enabling efficient processing of large-scale scRNA-seq data, but also exhibits a notable advantage in optimizing computational efficiency while maintaining high performance. The data comparison shown in Figure 5 further verifies the efficiency and practicality of scVGATAE among single-cell RNA-seq clustering methods. 

## 4. Conclusions

With the rapid advancements in single-cell RNA sequencing technology, a vast amount of high-dimensional, sparse, and complex gene expression data has been generated. Efficiently processing these data to extract biologically meaningful information, particularly through unsupervised learning methods for cell clustering, poses a significant challenge in the field of bioinformatics. To address this need, we have developed the scVGATAE model. The success of scVGATAE lies primarily in three aspects. Firstly, it utilizes NE to construct a high-confidence cell graph, effectively eliminating noisy edges in the cell graph. Secondly, by leveraging a variational graph attention autoencoder, it aggregates topological information from the cell network, fully exploiting relationships between cells and significantly enhancing model efficiency. Thirdly, scVGATAE adaptively selects appropriate model training epochs for different datasets, breaking away from the traditional qualitative mindset of manually tuning hyperparameters for neural network training and exhibiting superior generalization capabilities.

Our experimental results show that after removing noisy edges through the NE module, the cellular graph becomes more reliable, and scVGATAE effectively leverages the latent associations between cells to learn a more favorable low-dimensional embedding of cells, significantly boosting the clustering performance.

Despite the fact that scVGATAE outperforms existing methods in terms of efficiency and clustering performance, it still has some shortcomings. The number of cell clusters in scVGATAE is determined using the Leiden method, which may result in weaker generalization capabilities. Moreover, the performance of scVGATAE’s ability to adaptively select the appropriate number of model training epochs needs improvement. For instance, in the case of the Adipose dataset, the results obtained when using the adaptive model training epochs were not as good as those achieved through manual parameter tuning (e.g., setting the model training epochs to 180).

In the future, we will focus on enhancing the generalizability of identifying the optimal number of clusters for various datasets and explore other more effective clustering methods.

## Figures and Tables

**Figure 1 biology-13-00713-f001:**
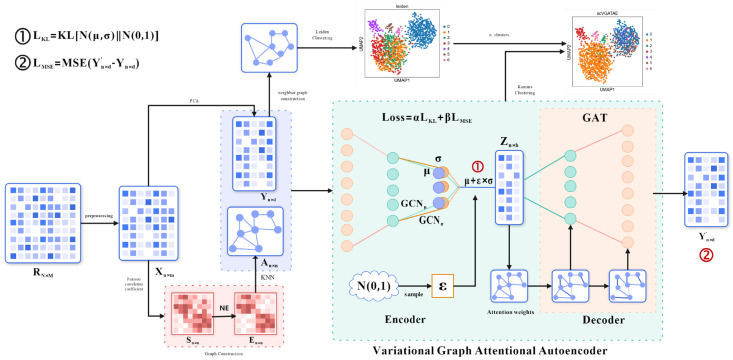
Architecture of scVGATAE. Firstly, the raw gene expression matrix $R_{N\times M}$ undergoes data preprocessing to obtain a preprocessed matrix $X_{n\times m}$. To obtain the two types of input data required for the variational graph attention autoencoder, namely the cell feature expression matrix $Y_{n\times d}$ and the cell graph $A_{n\times n}$, two separate operations are performed on the matrix $X_{n\times m}$. One operation involves applying PCA dimensionality reduction to the matrix $X_{n\times m}$ in order to reduce computational complexity. The other operation utilizes Pearson correlation coefficients to calculate a similarity matrix $S_{n\times n}$ between cells, which is then denoised through Network Enhancement (NE) to obtain the matrix $E_{n\times n}$. Based on the matrix $E_{n\times n}$, the k-nearest neighbors algorithm is executed, and the resulting matrix is normalized to ultimately derive the cell graph $A_{n\times n}$. Next, the cell feature expression matrix $Y_{n\times d}$ and the cell graph $A_{n\times n}$ are input into the variational graph attention autoencoder (VGATAE) model. The encoder, based on variational inference, learns the mean μ (represented by purple nodes) and variance σ (represented by yellow nodes) of the cells’ low-dimensional vector representations. Both μ and σ are learned through a two-layer graph convolutional network (GCN), sharing the first layer but having their own second-layer GCNs, ${GCN}_{\mu }$ and ${GCN}_{\sigma }$, respectively. To avoid sampling operations interfering with gradient descent, scVGATAE samples a value ε from the standard normal distribution $N\left(0,1\right)$ and then calculates the cells’ low-dimensional representation $Z_{n\times h}$ using the formula Z=μ+ε×σ. Subsequently, a decoder composed of two stacked graph attention networks reconstructs the cell feature expression matrix $Y_{n\times d}^{\prime }$. This encoding and decoding process is repeated until a stopping condition is met. Finally, the obtained low dimensional representation $Z_{n\times h}$ of cells is subjected to k-means clustering, and the number of cell clusters is calculated using the Leiden algorithm.

**Figure 2 biology-13-00713-f002:**
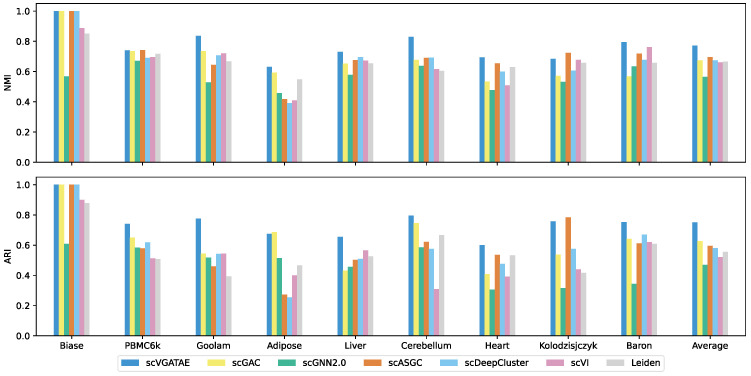
NMI and ARI scores of scVGATAE and six comparison methods in nine datasets, see also Appendix A.

**Figure 3 biology-13-00713-f003:**
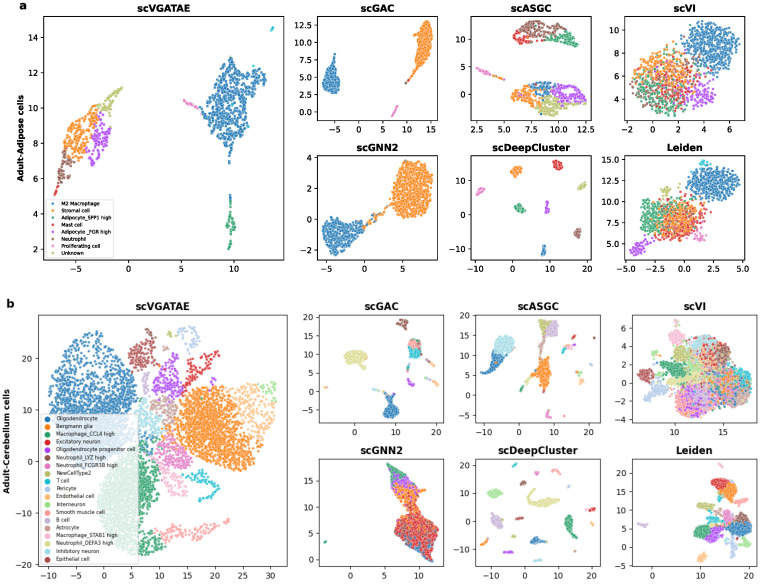
The visualization of the identified clusters of scVGATAE and six competitive methods are drawn. These points represent each sample cells, with different colors indicating different data labels. (**a**) The Adult-Adipose cell. (**b**) The Adult-Cerebellar cells. Where scGNN2 stands for scGNN2.0.

**Figure 4 biology-13-00713-f004:**
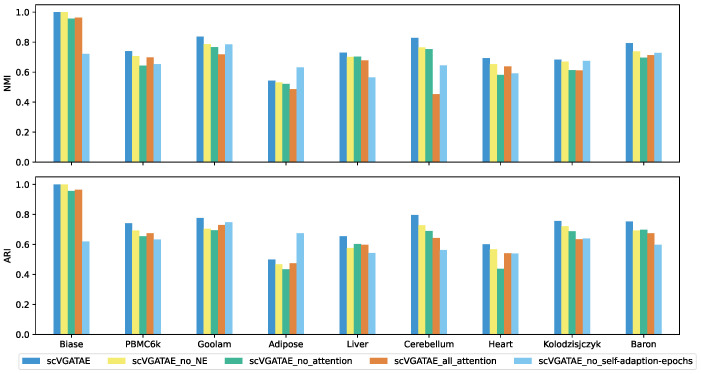
NMI and ARI comparison of scVGATAE and its four different ablation study subjects, see also Appendix A.

**Figure 5 biology-13-00713-f005:**
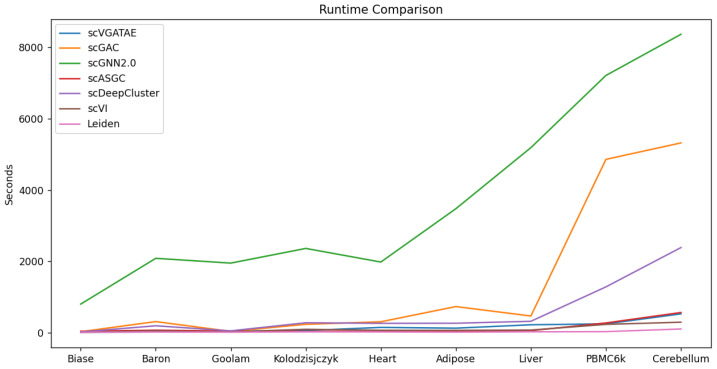
Comparison of the runtime of 7 methods on 9 datasets, see also Appendix A.

**Table 1 biology-13-00713-t001:** Details of the 9 evaluated datasets.

Datasets	Databases	Cells	Genes	Subtypes
Biase [24]	GEO	49	21489	3
PBMC6k [25]	GEO	5419	32738	10
Goolam [26]	GEO	124	41480	5
Adipose [23]	HCL	1372	12154	8
Liver [23]	HCL	1811	12631	16
Cerebellum [23]	HCL	7324	19244	19
Heart [23]	HCL	1308	12689	12
Kolodzisjczyk [27]	Array Express	704	10685	3
Baron [28]	GEO	822	14878	13

## Data Availability

https://github.com/duddubududu/scRNA-seq-data (accessed on 21 August 2024).

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
