# Peer review of "scVGATAE: A Variational Graph Attentional Autoencoder Model for Clustering Single-Cell RNA-seq Data"

_biology, 2024, doi:10.3390/biology13090713_

Round 1

Reviewer 1 Report

Comments and Suggestions for Authors This study introduces scVGATAE, an unsupervised clustering method for scRNA-seq data. The method aims to address challenges in clustering scRNA-seq data. scVGATAE presents a new approach for improving clustering performance in scRNA-seq data analysis, potentially contributing to better understanding of cell heterogeneity and biological processes at the single-cell level.

However, 

1. Typos, like "trajecues" in the Abstract. Please proofread carefully.

2. The code or package is not publicly available in a repository. Not sure how one could use scVGATAE for their own single cell data. 

3. What are the specific gaps in existing methods that scVGATAE addresses.

4. Figure 3, very hard to read and not sure how it shows that scVGATAE better than other methods.

5. All figures with low resolution. Figure 5 cannot see 7 lines. 

6. Table 1, HCL and GEO are not sequencing platforms, they are databases, please correct. Also, how the numbers of subtypes are defined. When I check citation [18] and see pbmc6k data, it says it has 4 cell types in [18] Table 1, but not 10 subtypes, what is the definition of subtypes. And if possible list the types from the original study for all the datasets used in the study. 

Comments on the Quality of English Language 1. Typos, like "trajecues" in the Abstract. Please proofread carefully. 2. Some words have hyphens, please correct the format. Like "com- pared“ in the 3.1 Compared with the other six methods, scVGATAE significantly improves clustering performance section.

Reviewer 2 Report

Comments and Suggestions for Authors

Xiaoyang Wu et al. present a new method called scVGATAE, which introduces a novel variational graph autoencoder model to learn low-dimensional representations of cells from single-cell RNA-seq datasets. After reviewing the manuscript, I have the following suggestions:

1.scvi-tools is the most widely used VAE-based method for single-cell RNA-seq analysis and should be compared with scVGATAE in the manuscript.

2 How many replicates were used for each dataset? Additionally, can scVGATAE reduce batch effects between samples from the same tissue?

3 The cell type labels on the UMAPs in Figure 3 are unclear; please provide high-resolution figures. Regarding the method scDeepCluster in Figure 3, the authors claim that it completely loses the correlation between the clusters. How was it determined which clusters should be closer than others?

4.Are there any hyperparameters in scVGATAE? Do the results vary when using different hyperparameter settings?

Reviewer 3 Report

Comments and Suggestions for Authors

Wu et.al proposed scVGATAE as a novel method for learning improved single cell RNAseq embeddings and identifying cell clusters. Several concerns needs to be addressed before it can be considered for publication.

1.        In main text, authors mentioned MSE and KL divergence are used to construct loss function. However, in Fig 1, MAE is labeled as part of the loss function, which is not consistent.

2.        The how are alpha and beta in loss function determined? I understand they can be empirical, but authors need to provide more details such as what procedures they use to choose alpha and beta, and metrics they are selected against.

3.        It seems like the learned cell embedding through a semi-supervised algorithm is general, and can be used beyond clustering. What other use cases are there? Can authors discuss about it?

Round 2

Reviewer 2 Report

Comments and Suggestions for Authors

The author addressed all my questions.

Author Response

Thank you sincerely for your thorough review . We are grateful for your dedication to ensuring the quality of our work.